# Explore the Way: Exploring Reasoning Path by Bridging Entities for Effective Cross-Document Relation Extraction

**Junyoung Son, Jinsung Kim, Jungwoo Lim, Yoonna Jang, Heuiseok Lim**[*]
Computer Science and Engineering, Korea University
Republic of Korea
{s0ny,jin62304,wjddn803,limhseok}@korea.ac.kr

## Abstract

Cross-document relation extraction (CodRED) task aims to infer the relation between two entities mentioned in different documents within a *reasoning path*. Previous studies have concentrated on merely capturing implicit relations between the entities. However, humans usually utilize explicit information chains such as hyperlinks or additional searches to find the relations between two entities. Inspired by this, we propose **P**ath w**I**th exp**LO**ra**T**ion (**PILOT**) that provides the enhanced reasoning path by exploring the explicit clue information within the documents. **PILOT** finds the bridging entities that directly guide the model with paths between the given entities and then employs them as stepstones to navigate desirable paths. We show that models with **PILOT** outperform the baselines in the CodRED task. Furthermore, we provide a variety of analyses to verify the validity of the reasoning paths constructed through **PILOT**, including evaluations using large language models such as ChatGPT.

## 1 Introduction

The relation extraction (RE) task aims to predict relations between two entities in a given text and is an essential basis for application tasks such as knowledge base construction (KBC) and question answering (QA) (Swampillai and Stevenson, 2010; Ji et al., 2010; Son et al., 2022). Although the sentence-level and document-level RE have achieved substantial performance (Zhang et al., 2017; Yao et al., 2019; Yang et al., 2021; Zhang et al., 2021), it is still challenging to identify the relations across multiple documents. Since the relations between the entities can be inferred from the multiple documents in the wild, cross-document relation extraction (CodRED) is suggested by Yao et al. (2021) which provides the relation between two entities (i.e., *head* and *tail*) across the multiple documents (Wu et al., 2023).[1]

---
[*]Corresponding author

[1]In the vast knowledge repository of Wikidata, more than 57.6% of the facts are not encapsulated within single

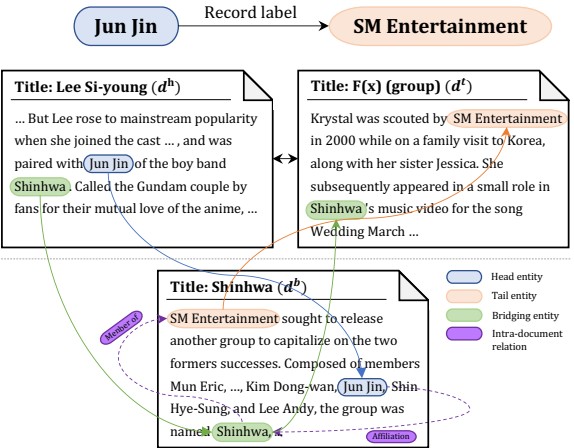

Figure 1: An illustrated example of $d^h$ and $d^t$ being linked by bridging entity *Shinhwa*. The previous reasoning path infers the relation between $h$ and $t$ by using only $d^h$ and $d^t$, but our method constructs the reasoning chain by utilizing the bridging entity *Shinhwa*.

To implement the cross-document RE model, Wang et al. (2022) has enhanced the method of capturing bridging entities by entity-based document-context filter and utilizes a cross-document entity relation attention module for enhanced connection awareness between the reasoning paths. On the contrary, Lu et al. (2022) focuses on extracting evidence path and ranking with dense retrieval without explicit consideration for the bridging entity. Likewise, previous studies have only attempted to model the bridging entities implicitly, neglecting to exploit the potentially valuable additional information that bridging entities can provide.

However, utilizing the explicit information relevant to bridging entities, which can provide a definitive guide in the exploration of reasoning paths, is crucial for identifying the correct relation. When inferring relations between entities using multiple documents, humans may not follow arbitrary in-

Wikipedia documents but are instead distributed across multiple documents.

formation. Instead, they instinctively draw upon explicitly linked information from the bridging entity between documents - that which connects a path to the tail entity and provides structural clues about the relation.

In this paper, we introduce **P**ath w**I**th exp**LO**ra**T**ion (**PILOT**), an effective way of constructing explicit and structural reasoning path utilizing bridging entities. According to **PILOT**, it first finds the bridging entities with structural entity information, such as intra-document relation, and retrieves the documents that are related to the bridging entities. Afterward, we rerank the documents by entity-aware scoring function and leverage the high-scored documents as the stepstones connecting the *head* and *tail*. The RE models with **PILOT** showed consistent improvement in performance compared to the baselines in the CodRED task. In addition, we provide an analysis with a large language model (LLM) such as ChatGPT (OpenAI-Blog, 2022) on whether our enhanced reasoning path leads to more knowledgeable and relevant information.

## 2 Method

Our proposed method revolves around leveraging the most significant bridging entities to construct a reasoning path between the *head* and *tail* entities. Initially, we employ structural information *(i.e., intra-document relations)* and the dense retriever to assemble a set of candidate bridging entities and their corresponding documents. Subsequently, we select the most relevant among these candidates through entity-based filtering and scoring methods. Finally, we utilize the final bridging entities to construct an expanded reasoning path.

### 2.1 Preliminaries

**Task Definition**    In cross-document RE, a *head* entity and a *tail* entity are denoted as $h$, and $t$, respectively. $P = \{p_i\}_{i=1}^N$ is a set of reasoning paths, and each reasoning path $p_i$ has two documents $(d_i^h, d_i^t)$, including $h$ and $t$. The task aims to infer the relation $r \in \mathcal{R}$ between two entities where $\mathcal{R}$ is a pre-defined relation set. If a particular entity is mentioned in both $d_i^h$ and $d_i^t$, it can consider this entity as a bridging entity between them. According to Yao et al. (2021), There are potential bridging entities that establish the relation between the documents (4.7 on average). We denote a set of the *bridging entities* as $E^b = \{e_i^b\}_{i=1}^M$ between two documents.

## 2.2 PILOT

### 2.2.1 Bridging Document Retrieval

**Entity-aware Retriever**    To find the relevant documents $D_e = \{d_e\}_{i=1}^L$ when given entity $e$ as a query, we train a dense retriever (Karpukhin et al., 2020). We first collect the entities which have their corresponding Wikipedia page from CodRED and Wikidata to construct the dataset for training the retriever. Afterward, we regard the Wikipedia page of the entity $e$ as a positive example $p^+$ and Wikipedia pages extracted from BM25 following Karpukhin et al. (2020) as the negative examples $\{p_j^-\}_{j=1}^J$. The training objective follows contrastive learning (Chen et al., 2020), as shown in Equation 1.

$$l(e, p^+, \{p_j\}_{j=1}^J) =$$
$$-\log \frac{\exp^{sim(e, p_i^+)}}{\exp^{sim(e, p_i^+)} + \sum_{j=1}^J \exp^{sim(e, p_j^-)}}. \quad (1)$$

**Bridging Candidate Construction**    To build a set of candidate bridging entities between two documents, we first utilize the shared entity set $E^{cand} = E^h \cup E^t$, where $(E^h, E^t)$ is the set of all entities that exist in $(d^h, d^t)$ respectively. With the dense retriever, we retrieve a set of documents $D^{cand} = \{d^{cand}\}_{i=1}^{|D^{cand}|}$ by using $e^{cand} \in E^{cand}$ as a query.

**Entity-based Filtering**    After retrieving the set of documents, we filter out the documents to preserve more relevant documents to the $e^{cand}$. In detail, we exclude $d^{cand}$ if $e^{cand}$ is not mentioned in $d^{cand}$. We further validate $e^{cand}$ exploiting structural information from Wikidata. If the $e^{cand}$ is connected to the $h$ and $t$ according to the Wikidata, we use them as a bridging entity candidate or, otherwise, filter out.

**Bridging Entity Scoring**    To quantify the relevance of candidate entities linked to the head and tail entities, we score each entity based on the given document. In detail, given an entity $e^{cand}$ and the document $d$, we find the entities in the $d$ and count the number of occurrences of $e^{cand}$ as follows:

$$N(e^{cand}, d) = \sum_{e \in E^d} \delta(e^{cand}, e), \quad (2)$$

where $\delta(e_1, e_2)$ returns 1 if $e_1 = e_2$ otherwise 0. Based on this score, we get the top-$k$ bridging entities $\hat{E}^b$ and employ them to construct a path.

| Method | Development | | | | Test | |
| --- | --- | --- | --- | --- | --- | --- |
| | AUC ($\sigma$) | F1 ($\sigma$) | P@500 ($\sigma$) | P@1000 ($\sigma$) | AUC ($\sigma$) | F1 ($\sigma$) |
| End-to-End (Yao et al., 2021) | 47.94 | 51.26 | 62.80 | 51.00 | 47.46 | 51.02 |
| + PILOT | 53.23 (0.59) | 56.12 (0.70) | 70.86 (0.92) | 55.57 (0.25) | 54.31 (1.23) | 57.33 (1.31) |
| + PILOT (- Wikidata) | 52.98 (0.98) | 55.72 (0.24) | 72.72 (0.80) | 55.60 (0.21) | 53.37 (0.53) | 57.52 (1.01) |
| ECRIM (Wang et al., 2022) | 60.91 | 61.12 | 78.89 | 60.17 | 60.67 | 62.48 |
| + PILOT | 63.83 (1.06) | 63.30 (0.35) | 79.48 (1.70) | 62.54 (0.35) | 62.90 (0.96) | 63.86 (1.01) |
| + PILOT (- Wikidata) | 63.31 (0.56) | 62.53 (0.41) | 78.54 (1.45) | 62.24 (0.61) | 62.19 (1.05) | 61.75 (0.74) |

Table 1: Comparisons of performances with the baselines on CodRED. ECRIM (Wang et al., 2022) is the existing state-of-the-art model. Our test results are obtained from the official website of CodRED on Codalab.

$$I(e^{cand}) = (N(e^{cand}, d^h) + N(h, d^{cand})) * \\ (N(e^{cand}, d^t) + N(t, d^{cand})), \quad (3)$$

Based on this score, we get the top-$k$ bridging entities $\hat{E}^b$ and employ them to construct a path.

### 2.2.2 Path Construction

Finally, we construct an expanded reasoning path $P'$ by employing the existing reasoning path $P = \{p_i\}_{i=1}^{|P|}$ and $\hat{E}_i^b = \{\hat{e}_{ij}^b\}_{j=1}^{|\hat{E}_i^b|}$. The detailed equation constructing $P'$ is as follows:

$$P' = \bigcup_{i=1}^{|P|} \bigcup_{j=1}^{|\hat{E}_i^b|} \{(d_i^h, d^{\hat{e}_{ij}^b}, d_i^t) \,|\, (d_i^h, d_i^t) \in P\}, \quad (4)$$

where $d^{\hat{e}_{ij}^b}$ is a document retrieved by Entity-aware retriever using query as $\hat{e}_{ij}^b$.

## 3 Experiments and Analysis

We apply **PILOT** to the existing reasoning path provided by (Yao et al., 2021) to construct an extended reasoning path. For the experiment, we utilize End-to-End (Yao et al., 2021) and ECRIM (Wang et al., 2022) models as baselines. The detailed experimental settings are described in 4.

### 3.1 Experimental Results in CodRED

We show the experimental results of **PILOT** method on the CodRED task in Table 1. These results demonstrated a consistent enhancement in the performance across all baselines when they were incorporated with **PILOT** method. In the End-to-End baseline, **PILOT** significantly improves performance across both the development and test sets. In particular, the substantial performance gain with 6.5% in the test set shows the generalization ability of our method. Even with the state-of-the-art model ECRIM, we observed a consistent improvement in

performance. Specifically, we witnessed accuracy improvements of up to 3.54% in the development set and 2.23% in the test set, validating the broad applicability of **PILOT**. When Wikidata filtering is removed, the performance decreases 0.4% for the End-to-End and 0.52% for the ECRIM in the development set, respectively. Especially in ECRIM, the f1 score drops 1.43% in the test set. These results suggest the central role of Wikidata filtering in **PILOT**, underlining the importance of structured information in navigating relational information for cross-document reasoning.

### 3.2 Experimental Results for Entity-aware Retriever

| Method | Development | | | Test | | |
| --- | --- | --- | --- | --- | --- | --- |
| | Hits@1 | R@5 | MRR | Hits@1 | R@5 | MRR |
| DR (dist) | 89.28 | 97.43 | 92.90 | 88.92 | 97.33 | 92.70 |
| DR (train) | 94.10 | 98.59 | 96.16 | 93.68 | 98.47 | 95.90 |
| DR (d. → t.) | **94.41** | **98.97** | **96.50** | **94.03** | **98.69** | **96.18** |

Table 2: Retrieval performances of Entity-aware Retriever on the development and test sets. Both 'dist' and 'd.' refer to the distant supervision manner.

The experimental results of the Entity-aware Retriever used for bridging document retrieval in **PILOT** are presented in Table 2. This process is significant since retrieving documents with low relevance to the entity could lead to a substantial performance decrease, creating a potential bottleneck. The Entity-aware Retriever, with an accuracy of 94.41% and 94.03%, hints at its pivotal role in PILOT, suggesting its ability to provide relevant bridging context to the bridging entities.

### 3.3 Evaluating the Capability to Distinguish Correct Reasoning Paths

We evaluate path-level accuracy rather than entity pair-level to validate the proposed method's capability to distinguish correct reasoning paths despite

extremely noisy settings. Because CodRED consists of only 6.7% of positive reasoning paths and the remaining 93.3% of N/A reasoning paths, we assume that the path-level evaluation can be used to estimate this ability. As shown in Table 3, we can

| Method | Path-level AUC |
| --- | --- |
| End-to-End (Yao et al., 2021) | 79.24 |
| + PILOT | **80.31** *(+1.07)* |

Table 3: Results of Path-level Accuracy on the development set.

observe that the path-level accuracy improves when the bridged reasoning paths expanded by PILOT are applied to the original reasoning paths despite their highly noised setting.

### 3.4 Quantity of Bridging Entities Explored

As shown in Figure 2, the RE performance initially improves with more bridging entities. However, as the number of these entities continues to rise, a performance drop is observed. This suggests that using too many bridging entities can introduce more noise, leading to decreased effectiveness. In addition, the performance greatly drops when the number of bridging entities is 5. This result reflects the fact of exceeding the average number of potential bridging entities (4.7 on average (Yao et al., 2021)).

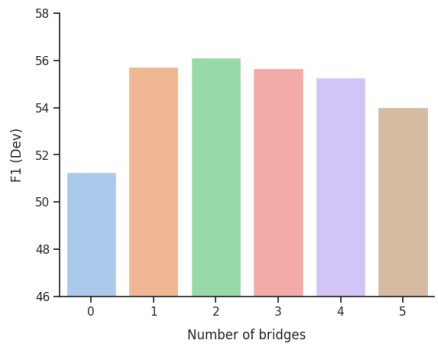

Figure 2: Results on F1 score with respect to the number of bridging entities provided per path.

### 3.5 Bridging Entity Type Selection

We demonstrate the cross-document RE performance based on the type of score function as in Figure 3. We compare four settings. '*w/o bridging*' is the setting where only $(d^h, d^t)$ is given as a reasoning path. '*All entities*' indicates the setting where the bridging entities are randomly selected from

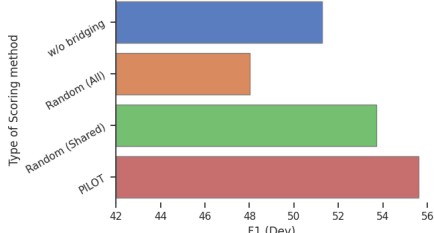

Figure 3: Achievement results on F1 by the type of scoring module to select the bridging entity.

the set of entities that appear in $(d^h, d^t)$. '*Shared entities*' is the setting where the bridging entities are randomly selected from the set of entities commonly appearing in $d^h$ and $d^t$.

As shown in Figure 3, the reasoning path constructed by **PILOT** shows the most outstanding performance. The performance of random entities ('*All entities*') is significantly worse than when the bridging entity is not utilized. It indicates that the performance improvement is not simply due to an increase in the amount of information in the bridging entity, but rather to the importance of finding genuinely relevant information.

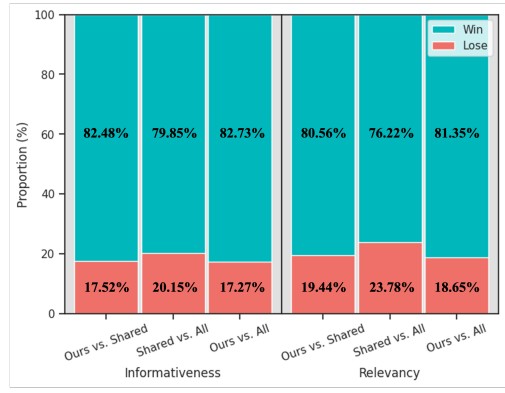

Figure 4: Win vs. Lose results (%) evaluated by Chat-GPT between the paths randomly chosen and construed by **PILOT**.

### 3.6 ChatGPT Evaluation

To show the efficacy of our reasoning path from **PILOT** method, we ask ChatGPT to evaluate between the cases where the path is explored by **PILOT** and the path is constructed with random entities.[2] The evaluation criteria are as follows: 1) **Informativeness** indicates how many clues the provided bridging entity information contains to infer a relation, and 2) **Relevancy** indicates how much relevant the

---

[2] The evaluation prompt template using ChatGPT is illustrated in Appendix 8.

given reasoning path, including the bridging entity, is with the given pair of entities (*head*, *tail*). Figure 4 illustrates that our reasoning paths got higher win proportion compared to both types of random paths by 64.96%p and 65.46%p, respectively, and this tendency is also similar in relevancy. In addition, *Ours vs. All* has a higher win rate than *Shared vs. All*. These results imply that our reasoning paths with **PILOT** method include more information for relation inference and act as the relevant evidence to explain the relation between *head* and *tail*.

### 3.7 Human Evaluation

|  | Informativeness | | | Relevancy | | |
|---|---|---|---|---|---|---|
|  | Win | Tie | Lose | Win | Tie | Lose |
| Ours vs. Shared | 55 | 28 | 7 | 53 | 30 | 7 |
| Ours vs. All | 61 | 27 | 2 | 59 | 29 | 2 |

Table 4: Results of Human Evaluation.

To support the results mentioned in the Chat-GPT Evaluation, we conducted a human evaluation by employing three individuals. More specifically, we presented each participant with 30 text snippets extracted from randomly sampled reasoning paths and had them undergo an AB test in comparison with the random baselines. As shown in Table 4, the results of the human evaluation exhibited a trend similar to that of the ChatGPT Evaluation. Notably, Our method demonstrated a win rate of 61.1% against Shared and 67.7% against All in Informativeness. In terms of Relevancy, it showed a win rate of 58.9% versus Shared and 65.5% against All. Additionally, the lower Lose rate in comparison to the Win and Tie ratios suggests that our methodology has significantly enhanced the quality of the reasoning path.

## 4 Conclusion

We proposed **PILOT**, which is the method that explores the reasoning path by utilizing the bridging entities and their documents for the CodRED task. Based on the filtering and scoring, **PILOT** exploits the explicit and structural information from metadata such as Wikidata, and constructs the enhanced reasoning path. In the experiments, the models with **PILOT** showed improvements compared to the baselines and outperformed the existing state-of-the-art model. Moreover, we provided additional experiments and extensive analyses to validate the efficacy of explored reasoning path and analysis.

## Limitations

Because **PILOT** constructs an expanded version of the reasoning path based on the given reasoning path which is a document pair $(d^h, d^t)$ for $h$ and $t$ entities, it requires an initial reasoning path to build an expanded one. This suggests that regardless of how well the bridging context is selected, the performance could be compromised if the quality of the initial reasoning path is not adequate, implying a dependency on the initial path. This factor should be taken into account when building paths in the open-settings in the future. In addition, the scoring function in **PILOT** is calculated using the number of mentions in each document, neglecting potentially important context which is infrequent but has a critical role between the entities. As Table 7 demonstrated, the bridging entities chosen from **PILOT** between entities can be nothing when the retrieved and filtered results return nothing. In other words, while this approach effectively filters out some irrelevant paths, it also carries the potential downside of losing valuable information that could contribute to inferring the relations between the entities.

## Acknowledgements

This research was supported by the MSIT(Ministry of Science and ICT), Korea, under the ITRC(Information Technology Research Center) support program(IITP-2023-2018-0-01405) supervised by the IITP(Institute for Information & Communications Technology Planning & Evaluation). This work was supported by Institute of Information & communications Technology Planning & Evaluation(IITP) grant funded by the Korea government(MSIT) (No. 2020-0-00368, A Neural-Symbolic Model for Knowledge Acquisition and Inference Techniques). Also, this research was supported by Basic Science Research Program through the National Research Foundation of Korea(NRF) funded by the Ministry of Education(NRF-2021R1A6A1A03045425).

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

# Appendix

## A   Experimental Setup

### A.1   Hyperparameters

We conduct our experiments using the closed-setting of CodRED. We employ the cased version of BERT-base model as the encoder. We use a learning rate of 3e-5 and apply a linear warm-up and learning rate schedule with the weight decay value 0.01. The batch size is 4. We train the baseline models on 4 NVIDIA RTX A6000 GPUs for 10 epochs.

### A.2   Implementation Details

To select a text snippet from the reasoning path, we follow the previous method (Yao et al., 2021), which extracts text snippets from the first mention of *head* and *tail* entities in each document. A slight difference from the previous method lies in our use of the bridging entity's document. In order to extract additional text snippets from the bridging entity's document as well, we augment the original text snippets by incorporating sections where the *head* and *tail* mentions first appear from the bridging entity's document.

### A.3   Evaluation Metrics

As evaluation metrics, we utilize F1/AUC/P@500/P@1000 for the development set and F1/AUC for the test set following the previous studies (Yao et al., 2021; Wang et al., 2022). All of the test results are obtained from the official website of CodRED on Codalab.

## A.4 Dataset Statistics

|  | Train | Dev | Test |
|---|---|---|---|
| # Positive facts | 2,733 | 1,010 | 1,012 |
| # N/A facts | 16,668 | 4,558 | 4,523 |
| # Bridges | 613,566 | 195,766 | 197,888 |
| # reasoning paths | 129,548 | 40,740 | 40,524 |

Table 5: Statistics of CodRED.

|  | Train | Train (dist.) | Dev | Test |
|---|---|---|---|---|
| # query-doc pairs | 9,042 | 95,310 | 4,601 | 4,576 |

Table 6: Statistics of the Entity-aware Retriever dataset.

| # has bridge | | # no bridge | |
|---|---|---|---|
| n/a | positive | n/a | positive |
| 40221 | | 519 | |
| 2553 | 37668 | 5 | 524 |

Table 7: Statistics of bridging entities for the constructed path using **PILOT**.

## B ChatGPT Evaluation Prompt Template

```
Task Instruction

Between the two cases, A and B, choose
the case where 1) the 'Text snippet' is more
informative for inferring the 'Relation' and
2) the case where the 'Relation' and the 'Text
snippet' are more relevant.
―――――――――――――――――――――――――――――――
Examples

Example 1:
Case A:
* Subject: Europa Plus
* Object: Soviet Union
* Relation: Europa Plus is country of Soviet
Union.
* Text snippet: The most titled volleyball
team in the Soviet Union and in Europe (CEV
Champions League) is VC CSKA Moscow.
―――
Case B:
* Subject: Past Masters
* Object: rock music
* Relation: Past Masters is genre of rock
music.
* Text snippet: In 2010, the official canon of
thirteen Beatles studio albums, Past Masters,
and the "Red" and "Blue" greatest-hits albums
were made available on iTunes. Reviews for Dr.
Feelgood have been highly positive. Critics
remarked the renewed energy and entertaining
values that permeate the album, bringing the
listeners "in a world of everlasting party",
where they "savored the joys of trashy,
unapologetically decadent fun.
―――

1) Informativeness: A
2) Relevancy: A

###
...
###
―――――――――――――――――――――――――――――――
Input
Case A:
* Relation: {subject_entity_a} is {relation_a}
of {object_entity_a}.
* Text snippet: {reasoning_text_A}
――-
Case B:
* Relation: {subject_entity_b} is {relation_b}
of {object_entity_b}.
* Text snippet: {reasoning_text_B}
――-
1) Informativeness:
2) Relevancy:
```

Table 8: Prompt Template for ChatGPT evaluation. We provide three gold examples in a few-shot manner.