# OpenReview forum: "Explore the Way: Exploring Reasoning Path by Bridging Entities for Effective Cross-Document Relation Extraction"
_EMNLP/2023/Conference — EMNLP 2023 Findings_

### Official Review · Reviewer_F5va · 2023-08-04

**Soundness:** 3

**Excitement:**

4: Strong: This paper deepens the understanding of some phenomenon or lowers the barriers to an existing research direction.

**Paper Topic And Main Contributions:**

The cross-document relation extraction task involves deducing relationships between head and tail entities across various documents. Unfortunately, prior studies have overlooked the utilization of hyperlinks or supplementary searches to acquire relations of entity pairs. Therefore, this paper propose PILOT that exploring the explicit clue information within the documents by utilizing bridging entities. Experiments on three datasets verify its effectiveness.






**Reasons To Accept:**

1. The paper proposes a coherent story.
2. The RE models with PILOT show performance improvement in comparison to the baselines within the CodRED task.

**Reasons To Reject:**

1. More baselines are needed.
2. The presentation of the CodRED dataset is missing.
3. I recommend verifying the effectiveness of PILOT on the more datasets.

**Reproducibility:**

4: Could mostly reproduce the results, but there may be some variation because of sample variance or minor variations in their interpretation of the protocol or method.

**Reviewer Confidence:**

3: Pretty sure, but there's a chance I missed something. Although I have a good feel for this area in general, I did not carefully check the paper's details, e.g., the math, experimental design, or novelty.

---

> ### Author Rebuttal · Authors · 2023-08-28
>
> Thanks for your constructive comments. We sincerely appreciate your time in reading the paper. Please find the responses below.
>
> &nbsp;
>
>     Reasons To Reject: More baselines are needed.
>
> Re: We genuinely appreciate your valuable suggestion. To the best of our knowledge, regarding CodRED, we have employed all the available baselines (End-to-End[1] and ECRIM[2]) for our evaluations. Given that CodRED itself is not a task that has been in existence for an extended period, the availability of additional potential alternatives is limited.
>
>
> &nbsp;
>
>     Reasons To Reject: The presentation of the CodRED dataset is missing.
>
> Re: We value your feedback concerning the presentation of the CodRED dataset. We aimed to convey the fundamental schema of the reasoning path in CodRED through Figure 1. Moreover, we have included dataset statistics in the Appendix. We apologize if the explanation for enhancing the understanding of the dataset appeared insufficient.
>
> We will provide detailed explanations of the CodRED dataset in the final version.
>
> &nbsp;
>
>     Reasons To Reject: I recommend verifying the effectiveness of PILOT on the more datasets.
>
> Re: We greatly appreciate your review and suggestion. Given that the CodRED task is designed to simulate a unique scenario of cross-document relation extraction using the reasoning path ceocept, there are no supplementary datasets available for method validation.
>
> Nevertheless, as you have rightly pointed out, we will certainly consider exploring the scalability of our approach to other multi-document tasks as part of our future research endeavors.
>
> &nbsp;
>
> References
>
> [1] Yao, Y., Du, J., Lin, Y., Li, P., Liu, Z., Zhou, J., & Sun, M. (2021, November). CodRED: A cross-document relation extraction dataset for acquiring knowledge in the wild. *In Proceedings of the 2021 Conference on Empirical Methods in Natural Language Processing*.
>
> [2] Wang, F., Li, F., Fei, H., Li, J., Wu, S., Su, F., ... & Cai, B. (2022, December). Entity-centered Cross-document Relation Extraction. *In Proceedings of the 2022 Conference on Empirical Methods in Natural Language Processing*

---

### Official Review · Reviewer_18D4 · 2023-08-04

**Soundness:** 3

**Excitement:**

3: Ambivalent: It has merits (e.g., it reports state-of-the-art results, the idea is nice), but there are key weaknesses (e.g., it describes incremental work), and it can significantly benefit from another round of revision. However, I won't object to accepting it if my co-reviewers champion it.

**Missing References:**

-

**Paper Topic And Main Contributions:**

The authors introduce an enhanced method for cross-document relation extraction, termed PILOT. This technique aims to discern relationships between two entities across multiple documents. Unlike traditional methods, PILOT does not solely rely on information from documents where the head (h) and tail (t) entities appear, but instead utilizes so called 'bridging entities' that frequently cooccur with head and tail to form an extended reasoning path across multiple documents. Relevant documents containing these bridging entities are sourced using a dense retriever, trained on Wikipedia data with contrastive learning. The retrieved documents are then re-ranked to pinpoint those most relevant for discerning the relationship between head and tail entities. Documents lacking the bridge entity are excluded, as are bridging entities without Wikipedia structural information linking it to both entities. After re-ranking, the best bridge entity candidates are used to expand the existing reasoning path. Experiments demonstrae that existing approaches combined with PILOT achieve better performance as measured by the CodRED benchmark dataset.

**Questions For The Authors:**

A: Concerning the ChatGPT evaluation in Section 3.5: Did you conduct experiments to ensure the model's assessment of Informativeness and Relevancy matches your definition? Could you provide a baseline comparison between the “shared” and “all” approaches, or against random text?

B: You mention using a dense retriever trained on Wikipedia articles, which retrieves documents related to a specific bridge entity, but discards those not directly mentioning the entity. Further documents are discarded if the wikipedia articles does not indicate a connection. One would expect that such strict filtering negates the benefits of a dense retriever. The description in the paper implies that entity mentions must exactly match the bridge entity. Could you please clarify how this works and whether entity disambiguation is used to identify instances that an exact match approach might miss?

**Reasons To Accept:**

* Task relevance
The considered cross-domain relation extraction task is challenging and timely

* Comprehensive model description
The path construction process is described in detail and the underlying reasoning is provided.

* Evaluation against SOTA baselines
PILOT is evaluated using two current state-of-the-art approaches, demonstrating an improvement in performance in both cases

**Reasons To Reject:**

* Modeling ambiguity
The authors conflate the concepts of entity and an entity's wikipedia/wikidata text, which is questionable from a modeling perspective. As a result, it is also unclear how generalizable the proposed method is to open world relation extraction. While the concept of finding intermediate entities to identify further relaion paths between entities is interesting, it is not obvious to which degree this is relevant in practice. For example, consider Figure 1: while entities dh and dt are related, this relation is evident from the *text* of db - this does not mean that db has to be a linking entity, though, as it could be an arbitrary non-entity text. At the very least, the authors should clean up the conceptual approach an make clear how much of their model is specific to the wikidata/wikipedia ecosystem.

* Evaluation
The authors repeteadly mention the significance of their results, with no evidence of statistical testing. Some results (e.g. Figure 4) are very close and require significance testing to support the author's claims.

* "Manual" LLM evaluation
The authors use ChatGPT prompting instead of a human error analysis, which seems questionable in its efficacy. In particular, the authors note that ChatGPT favoured the extracted relationship description produced by this paper's method over the baseline by approximately 65%. This outcome was determined by presenting ChatGPT with two relationship descriptions and some task examples, asking the model to choose the more "informative" and "relevant" option. However, as ChatGPT's performance in such evaluations is still under investigation and prompt design will heavily influence results, alternative evaluation methods should be used. While LLMs like ChaGPT might be useful in method design, they should not be expected to replace human evaluation quie yet.

**Reproducibility:**

4: Could mostly reproduce the results, but there may be some variation because of sample variance or minor variations in their interpretation of the protocol or method.

**Reviewer Confidence:**

4: Quite sure. I tried to check the important points carefully. It's unlikely, though conceivable, that I missed something that should affect my ratings.

**Typos Grammar Style And Presentation Improvements:**

The paper is overall well written, but contains numerous typos that should be corrected
* line 035: Remove the extra “the.”
* line 113-115: The sentence regarding how CodRED and Wikipedia were used to create the dataset for training the dense retriever is rather unclear.
* line 123: It seems a “-” superscript might be missing from p_j on the equation's left side.
* line 147: Equation 2 seems redundant, given the prior description about "counting the number of occurrences."
* Figure 3: Consider rephrasing “Achievement results on F1” to “Performance measured by F1 score for each type of …”

Further remarks:
* The paper would benefit from a more detailed description of the key characteristics of the dataset "CoDRED".
* Section 3.5 would benefit from a description of the extraction process for text snippets from the identified documents
* In Table 1, why are only precision values only given for P@500 and P@1000 and not much smaller values?

---

> ### Author Rebuttal · Authors · 2023-08-28
>
> We deeply thank the reviewer for their _insightful comment_ and detailed feedback on _evaluation_, _typos_, and _presentation improvements_. They are exceedingly helpful for us to improve our paper. Please find our responses in the following.
>
> &nbsp;
>
>     Reasons To Reject: Modeling ambiguity The authors conflate the concepts of entity and an entity's wikipedia/wikidata text, which is questionable from a modeling perspective. As a result, it is also unclear how generalizable the proposed method is to open world relation extraction. While the concept of finding intermediate entities to identify further relaion paths between entities is interesting, it is not obvious to which degree this is relevant in practice. For example, consider Figure 1: while entities dh and dt are related, this relation is evident from the text of db - this does not mean that db has to be a linking entity, though, as it could be an arbitrary non-entity text. At the very least, the authors should clean up the conceptual approach an make clear how much of their model is specific to the wikidata/wikipedia ecosystem.
>
>
> Re: To clarify our method, we first note that our main idea is to reflect human's search process into the reasoning path and the importance of utilizing entity-centric information such as hyperlinks (Linking Entity) to identify helpful bridging entities in the cross-document. As you mentioned, in the real world, there are many non-entity texts along with erroneous (also misaligned) entities which could be helpful to infer the relation.
>
> However, our study begins with the assumption that effective reasoning paths can be constructed by following explicit and structural information, such as hyperlinks, much like the process humans employ during their searches. To put it differently, we drew inspiration from the fact that people often reach their objectives more efficiently by utilizing information like hyperlinks rather than relying solely on a single search.
>
> Definitely, considering this kind of entity without structural information could indeed contribute to enhancing the quality of reasoning paths.
> However, this specific perspective wasn't the primary focus of our current study. We'd like to clarify that it's something we could explore in the future, especially in the more "in the wild" environments you suggested. We greatly appreciate your input on this idea.
>
> Furthermore, research involving entity-link information isn't limited to the domains of Wikipedia/Wikidata. By integrating existing systems like entity-linking and Named Entity Recognition systems into our method or developing a framework, we believe our approach could be extended to numerous other domains as well.
>
> In conclusion, while our study is not focused on addressing situations lacking structured information like hyperlinks, we hope you'll consider our motivation to effectively construct reasoning paths by leaveraging such explicit and structural information.
>
> &nbsp;
>
>     Reasons To Reject: Evaluation The authors repeteadly mention the significance of their results, with no evidence of statistical testing. Some results (e.g. Figure 4) are very close and require significance testing to support the author's claims.
>
>
> Re: We sincerely appreciate the constructive feedback. We acknowledge the importance of statistical validation to support our claims, particularly in Figure 4 (Section 3.5). Based on your suggestions, we have taken steps to include statistical testing results in that section to bolster our claims.
>
> For the experimental setting, we performed 7 different random samples and analyzed them with different random seeds.
>
> |                 |      Informativeness     |    |                           |   | Relevancy |    |                           |
> |:---------------:|:------------------------:|:--:|:-------------------------:|---|:---------:|:--:|:-------------------------:|
> |    **Method**   |         Win rate         |  | Confidence Interval (95%) |   |  Win rate |  | Confidence Interval (95%) |
> | Ours vs. Shared |          83.18%          |    |      81.88% ~ 84.48%      |   |   81.56%  |    |      80.22% ~ 84.48%      |
> |   Ours vs. All  |          82.66%          |    |      81.37% ~ 83.95%      |   |   81.45%  |    |      80.12% ~ 82.78%      |
> |  Shared vs. All | 79.85% (shared win rate) |    |      78.42% ~ 81.29%      |   |   76.22%  |    |      74.70% ~ 77.74%      |
>
>
>
> In addition, we will revise our paper to include all of statistical validation in the final version. For example, we provide additional statistical results into the Table 1.
>
> | **Method**     |                        | **Dev**        |                |                |                | **Test**       |                |
> |----------------|------------------------|----------------|----------------|----------------|----------------|----------------|----------------|
> |                |                        | **AUC**        | **F1**         | **P@500**      | **P@1000**     | **AUC**        | **F1**         |
> | **End-to-End** |                        | 47.94          | 51.26          | 62.80          | 51.00          | 47.46          | 51.02          |
> |                | **+PILOT**             | 53.23 (± 0.59) | 56.12 (± 0.7)  | 70.86 (± 0.92) | 55.57 (± 0.25) | 54.31 (± 1.23) | 57.33 (± 1.31) |
> |                | **+PILOT (-Wikidata)** | 52.98 (± 0.98) | 55.72 (± 0.24) | 72.72 (± 0.8)  | 55.60 (± 0.21) | 53.37 (± 0.53) | 57.52 (± 1.01) |
>
>
> &nbsp;
>
>     Reasons To Reject: "Manual" LLM evaluation The authors use ChatGPT prompting instead of a human error analysis, which seems questionable in its efficacy. In particular, the authors note that ChatGPT favoured the extracted relationship description produced by this paper's method over the baseline by approximately 65%. This outcome was determined by presenting ChatGPT with two relationship descriptions and some task examples, asking the model to choose the more "informative" and "relevant" option. However, as ChatGPT's performance in such evaluations is still under investigation and prompt design will heavily influence results, alternative evaluation methods should be used. While LLMs like ChaGPT might be useful in method design, they should not be expected to replace human evaluation quie yet.
>
> Re: To address this issue, we also conducted a human evaluation using 30 samples with 3 people, as shown in the table below:
>
> |            |     | Informativeness |      |     | Relevancy |      |
> |:----------:|:---:|:---------------:|:----:|:---:|:---------:|:----:|
> |            | Win |       Tie       | Lose | Win |    Tie    | Lose |
> | vs. shared |  55 |        28       |   7  | 53  | 30        | 7    |
> | vs. all    |  61 |        27       |   2  | 59  | 29        | 2    |
>
> According to our human evaluation results, participants expressed a preference for PILOT's reasoning path in terms of Informativeness (61.1% for vs. shared and 67.7% for vs. all) and Relevancy (58.9% for vs. shared and 65.5% for vs. all).
> We observed that these tendencies align with the results in the ChatGPT evaluation.
>
> Furthermore, while there is ongoing debate about the reliability of automated evaluations using LLMs such as ChatGPT, many studies do assert their validity [1, 2]. Particularly, in our study, entrusting the assessment of Informativeness and Relevancy to ChatGPT was motivated by the inspiration that our approach remains within the scope of research aligned with these findings.
>
> As you've advised, we certainly do not disregard the importance of human evaluation. We are well aware of its significance. Therefore, we are conducting evaluations on a larger scale for Human Evaluation, and we intend to incorporate these evaluation results into the final version of this work.
>
> Finally, we will ensure to provide comprehensive explanations in the final copy regarding the guidelines, results analysis, and evaluation environment for human evaluation.
>
>
> &nbsp;
>
>     Questions For The Authors A: Concerning the ChatGPT evaluation in Section 3.5: Did you conduct experiments to ensure the model's assessment of Informativeness and Relevancy matches your definition? Could you provide a baseline comparison between the “shared” and “all” approaches, or against random text?
>
> Re: We appreciate your question, which focuses on the quality of the ChatGPT evaluation in Section 3.5. As we mentioned in our previous response, we conducted a human evaluation that provides support for the assessment conducted through ChatGPT.
>
> In addition, as you requested, we provide a baseline comparison between the "shared" and "all" approaches using same criteria.
>
> |                 |      Informativeness     |    |                           |   | Relevancy |    |                           |
> |:---------------:|:------------------------:|:--:|:-------------------------:|---|:---------:|:--:|:-------------------------:|
> |    **Method**   |         Win rate         |  | Confidence Interval (95%) |   |  Win rate |  | Confidence Interval (95%) |
> |  Shared vs. All | 79.85% (shared win rate) |    |      78.42% ~ 81.29%      |   |   76.22%  |    |      74.70% ~ 77.74%      |
>
> This result shows that PILOT is preferred by approximately 2.8%p than Shared baseline in the setting of comparison with the All baseline.
>
> &nbsp;
>
>     Questions For The Authors B: You mention using a dense retriever trained on Wikipedia articles, which retrieves documents related to a specific bridge entity, but discards those not directly mentioning the entity. Further documents are discarded if the wikipedia articles does not indicate a connection. One would expect that such strict filtering negates the benefits of a dense retriever. The description in the paper implies that entity mentions must exactly match the bridge entity. Could you please clarify how this works and whether entity disambiguation is used to identify instances that an exact match approach might miss?
>
> Re: We appriciate your detailed question for our study. We used strict filtering at the entity level in our study because it aligns with how human search processes often begin with specific queries and then expand to consider entities or keyword-based units. This emphasizes the importance of initial searches followed by utilizing structural information like hyperlinks.
>
> Just like your concern, we also worried that strict filtering might cause information loss. In initial experiments, we attempted to use surface-level information like string matching for filtering instead of entity id. However, we found that this approach introduced more noise than the potentially useful information it could provide for constructing reasoning paths. The cost of building reasoning paths seemed to outweigh the benefits.
>
> Additionally, the CodRED dataset utilizes Entity ID from Wikidata (e.g., Q985235 for Fridley) for entity linking [3]. They employed a BERT-Large model trained on DocRED for Named Entity Recognition. Then, they aligned the recognized entity mentions with Wikidata's metadata like names and aliases for linking purposes.
>
> In summary, our focus was on capturing the efficiency of constructing reasoning paths by combining dense retrieval with structural information like hyperlinks, akin to how humans often proceed with searches. While non-link-related entity information does exist, we hope our motivation behind the synergy of dense retrieval and structural information, as well as our key results, shines through in this context.
>
>
> &nbsp;
>
> References
>
> [1] Chiang, C. H., & Lee, H. Y. (2023). Can Large Language Models Be an Alternative to Human Evaluations?. *In Proceedings of the 2023 Conference on Association for Computational Linguistics*.
>
> [2] Liu, Y., Iter, D., Xu, Y., Wang, S., Xu, R., & Zhu, C. (2023). Gpteval: Nlg evaluation using gpt-4 with better human alignment. *arXiv preprint arXiv:2303.16634*.
>
> [3] Yao, Y., Du, J., Lin, Y., Li, P., Liu, Z., Zhou, J., & Sun, M. (2021, November). CodRED: A cross-document relation extraction dataset for acquiring knowledge in the wild. *In Proceedings of the 2021 Conference on Empirical Methods in Natural Language Processing*.

---

### Official Review · Reviewer_shPs · 2023-08-12

**Soundness:** 3

**Excitement:**

3: Ambivalent: It has merits (e.g., it reports state-of-the-art results, the idea is nice), but there are key weaknesses (e.g., it describes incremental work), and it can significantly benefit from another round of revision. However, I won't object to accepting it if my co-reviewers champion it.

**Paper Topic And Main Contributions:**

In this paper the authors introduce an effective way of constructing explicit and structural reasoning path utilizing bridging entities, named PILOT. PILOT first finds the bridging entities with structural entity information, and retrieves the documents that are related to the bridging entities. Afterward, it reranks the documents by entity-aware scoring function and leverage the high-scored documents as the stepstones connecting the head and tail. PILOT achieved consistent improvement in performance compared to the baselines in the CodRED task. In addition, the authors provide an analysis with a large language model including ChatGPT on whether PILOT leads to more knowledgeable and relevant information.

**Reasons To Accept:**

1. Present an effective way of constructing explicit and structural reasoning path utilizing bridging entities for effective cross-document relation extraction.
2. Provide an analysis with a large language model including ChatGPT on whether PILOT leads to more knowledgeable and relevant information.
3. Achieved consistent improvement in performance compared to the baselines in the CodRED task.

**Reasons To Reject:**

1. The performance of PILOT is deeply dependent on the quality of the initial reasoning path.
2. The proposed mothed neglect potentially important context when calculating the function score using the number of mentions in each document.


**Reproducibility:**

5: Could easily reproduce the results.

**Reviewer Confidence:**

4: Quite sure. I tried to check the important points carefully. It's unlikely, though conceivable, that I missed something that should affect my ratings.

---

> ### Author Rebuttal · Authors · 2023-08-28
>
> Thanks for your constructive comments and suggestions. We sincerely appreciate your time reading the paper, and our responses to your comments are below.
>
> &nbsp;
>
>     Reasons To Reject: The performance of PILOT is deeply dependent on the quality of the initial reasoning path.
>
> Re: Because CodRED dataset has only 6.7% of positive reasoning paths and the remaining 93.3% of n/a reasoning paths, it can be said that PILOT can be employed to identify informative reasoning paths out of noises.
>
> Based on your opinion, we performed additional analysis to validate the efficacy of our method by investigating our results focused on the extremely noisy entity pairs that have more than 5 n/a paths with only 1~2 positive paths from the development set..
>
> |Method|Path-level accuracy|
> |---|---|
> |End-to-End|79.24|
> |+ PILOT|**80.31**|
>
> We observed that our method obtains the higher path-level accuracy, which implies the higher capability to identify an informative reasoning path despite the noise.
>
> Lastly, we agree that open-setting of CodRED, which requires constructing initial reasoning paths from scratch and also needing to reason the relations, is an important research field. However, this topic falls outside the scope of this paper, and we will leave this content for further researchers to explore.
>
> &nbsp;
>
>     Reasons To Reject: The proposed mothed neglect potentially important context when calculating the function score using the number of mentions in each document.
>
> Re: our main idea is about modeling how people search for information in the web to the process of reasoning path construction. When we search, we start with a query that represents what we want to find, and then we look at results based on the entity related to intention. This is similar to how PILOT works. So, in our approach, it's important to see the scoring method as part of a framework that combines with the dense retrieval. We are committed to providing a more comprehensive explanation of our approach in the revised manuscript.
>
> In addition, it's worth noting that several studies have demonstrated the effectiveness of combining the dense representation with the sparse representation [1][2].
>
>
> &nbsp;
>
> References
>
> [1] Glass, M., Rossiello, G., Chowdhury, M. F. M., Naik, A., Cai, P., & Gliozzo, A. (2022, July). Re2G: Retrieve, Rerank, Generate. *In Proceedings of the 2022 Conference of the North American Chapter of the Association for Computational Linguistics: Human Language Technologies*.
>
> [2] Dhingra, B., Zaheer, M., Balachandran, V., Neubig, G., Salakhutdinov, R., & Cohen, W. W. (2019, September). Differentiable Reasoning over a Virtual Knowledge Base. *In International Conference on Learning Representations.*

---

### Meta-Review · Area_Chair_iMjd · 2023-09-19

**Recommendation:** 3

**Metareview:**

The reviews have a consensus on slightly positive soundness and excitement.

Regarding soundness, the empirical observations show the proposed method improves two state-of-the-art baselines in the CodRED task, demonstrating the effectiveness of the proposed method.  The technical description on the path construction process is comprehensive and clear.

With respect to excitement, the idea of employing bridging entities for reasoning paths is novel.  However, its practical impact is obscure.

---

### Decision · Program_Chairs · 2023-10-07

**Decision:**

Accept-Findings

**Comment:**

The reviews have a consensus on slightly positive soundness and excitement.

Regarding soundness, the empirical observations show the proposed method improves two state-of-the-art baselines in the CodRED task, demonstrating the effectiveness of the proposed method.  The technical description on the path construction process is comprehensive and clear.

With respect to excitement, the idea of employing bridging entities for reasoning paths is novel.  However, its practical impact is obscure.